# The Impact of COVID-19 Pandemic on Inequity in Routine Childhood Vaccination Coverage: A Systematic Review

**DOI:** 10.3390/vaccines10071013

**Published:** 2022-06-24

**Authors:** Nicholas Spencer, Wolfgang Markham, Samantha Johnson, Emmanuelle Arpin, Rita Nathawad, Geir Gunnlaugsson, Nusrat Homaira, Maria Lucia Mesa Rubio, Catalina Jaime Trujillo

**Affiliations:** 1Division of Health Sciences, Warwick Medical School, University of Warwick, Coventry CV4 9JD, UK; wolfgang.markham@warwick.ac.uk; 2University of Warwick Library, University of Warwick, Coventry CV4 7AL, UK; samantha.johnson@warwick.ac.uk; 3Dalla Lana School of Public Health, University of Toronto, Toronto, ON M5T 3M7, Canada; emmanuelle.arpin@mail.untoronto.ca; 4Department of Pediatrics, Division of Community and Societal Pediatrics, College of Medicine-Jacksonville, University of Florida, FL 32209, USA; rita.nathawad@jax.ufl.edu; 5Faculty of Sociology, Anthropology, and Folkloristics, University of Iceland, IS-102 Reykjavik, Iceland; geir.gunnlaugsson@hi.is; 6Discipline of Paediatrics, University of New South Wales, Sydney, NSW 2031, Australia; n.homaira@unsw.edu.au; 7Pediatric Department, School of Medicine, Los Andes University, Cra 1 Nº 18A-12, Bogota 111711, Colombia; marialuciamesa@gmail.com (M.L.M.R.); catalinajaime.trujillo@gmail.com (C.J.T.)

**Keywords:** COVID-19, routine childhood vaccination coverage, inequity, systematic review

## Abstract

Background: Routine childhood vaccination coverage rates fell in many countries during the COVID-19 pandemic, but the impact of inequity on coverage is unknown. Methods: We synthesised evidence on inequities in routine childhood vaccination coverage (PROSPERO, CRD 42021257431). Studies reporting empirical data on routine vaccination coverage in children 0–18 years old during the COVID-19 pandemic by equity stratifiers were systematically reviewed. Nine electronic databases were searched between 1 January 2020 and 18 January 2022. The risk of bias was assessed using the Newcastle-Ottawa Quality Assessment Tool for Cohort Studies. Overall, 91 of 1453 studies were selected for full paper review, and thirteen met the inclusion criteria. Results: The narrative synthesis found moderate evidence for inequity in reducing the vaccination coverage of children during COVID-19 lockdowns and moderately strong evidence for an increase in inequity compared with pre-pandemic months (before March 2020). Two studies reported higher rates of inequity among children aged less than one year, and one showed higher inequity rates in middle- compared with high-income countries. Conclusions: Evidence from a limited number of studies shows the effect of the pandemic on vaccine coverage inequity. Research from more countries is required to assess the global effect on inequity in coverage.

## 1. Introduction

The measures taken by governments to suppress the COVID-19 pandemic, including restrictions on movements and lockdowns, have resulted in a reduction in routine childhood vaccination coverage in countries worldwide [1]. It is estimated that from January to December 2020, 30.0 million (27.6–33.1) children missed the third dose of diphtheria, tetanus and pertussis vaccine (DTP3), and 27.2 million (23.4–32.5) children missed the first dose of measles-containing vaccine (MCV1). Although coverage rates have recovered in many countries, millions of children remain unprotected against diseases such as measles [2], and recent improvements in vaccination coverage are in jeopardy [3].

Prior to the COVID-19 pandemic, inequity in routine vaccination coverage, although reducing, was reported, particularly in low- and middle-income countries (LMICs), with children in the poorest households and remote rural areas less likely to receive optimal vaccination coverage due to inadequate health infrastructure and supply chain problems [4]. The COVID-19 pandemic, similarly to earlier pandemics, is known to have the most significant impact on the most disadvantaged households and social groups [5,6], disrupting the already inadequate and overstretched health infrastructure in LMICs and high-income countries (HICs).

Equity in routine childhood vaccination coverage, especially for poor, marginalised and rural communities, was identified as essential to attaining Sustainable Development Goal 3.B.1 [3,7]. Inequity in vaccination access violates the child’s right to survival and optimal healthcare (United Nations Convention on the Rights of the Child Articles 6 and 24) [8], and the pandemic may have exacerbated this inequity. To explore the potential impact of pandemics and epidemics, including COVID-19, on vaccination coverage inequity, we undertook a systematic review of the literature from 1900 through to 2 June 2020; however, despite a comprehensive search strategy, no papers fulfilling the inclusion criteria were identified [9]. After our search end date, a research letter was published with data on inequity in routine childhood vaccination coverage in Karachi, Pakistan [10], prompting us to conduct a further search. Therefore, this systematic review aimed to address the existing gap in the body of evidence and assess the impact of the COVID-19 pandemic on inequities in routine childhood vaccination coverage.

## 2. Methods

The search strategy for published and pre-print literature was conducted by a health sciences librarian at the University of Warwick Library (SJ) to identify the published studies that reported data on inequity in routine childhood vaccination coverage during the COVID-19 pandemic. The review protocol was registered with PROSPERO (CRD 42021257431).

### 2.1. Literature Search

Eight online databases were searched, including MEDLINE, EMBASE, Web of Science, Cochrane Central Register of Controlled Trials (CENTRAL), Cochrane CDSR, Sociological Abstracts, ASSIA, and MEDRxiv (initial search only as indexed in PubMed from February 2020), with no language restrictions, initially from 1 January 2020 to 30 April 2021 and updated from 1 May to 18 January 2022. The WHO Global Research on Corona Virus (COVID19) was also consulted (https://www.who.int/emergencies/diseases/novel-coronavirus-2019/global-research-on-novel-coronavirus-2019-ncov accessed initially on 30 April 2021 and updated on 18 January 2022). Full details of the search strategies for each database are available in Appendix A. The following exploded MeSH terms were used in the search: Coronavirus or Coronavirus Infections; Vaccines/or vaccin*.mp. or exp Vaccination or Immunisation Programs/or exp Immunisation/or immunisation.mp.; inequality.mp. or exp Socioeconomic Factors or Poverty/or poverty.mp. or healthcare disparities.mp. or exp Healthcare Disparities or Income/or income.mp. or Social Class/or social status.mp. or social class.mp. or Educational Status/or educat*.mp. or Health Status Disparities/or exp Health Status or Health Services Accessibility; Child/or child *.mp. or infant *.mp. or exp Infant.

Studies were included in the review if they reported data by equity stratifiers on routine vaccination coverage in child populations (0–18 years) within or between HICs and LMICs during the COVID-19 pandemic. Equity stratifiers were defined as individual or area characteristics representing socioeconomic factors, wealth, indigenous status, or racial/ethnic identity that can identify population subgroups at risk of suffering from healthcare disparities [11].

Excluded studies included those that reported on adults in which data on children were not considered separately, inequity in vaccination coverage but not related to the COVID-19 pandemic and vaccination coverage during the COVID-19 pandemic without measures of inequity.

### 2.2. Study Selection

The records identified by the search were deduplicated. The initial screening of the papers for eligibility criteria and selection of studies for full paper review was conducted in duplicate by four pairs of researchers (NS and WM; RN and NS; GG and NH; MLM and CJ) working independently. Studies selected for full paper review were allocated equally to the same four pairs of researchers who independently assessed whether studies met the inclusion criteria. Disagreements were resolved by consensus and, where necessary, reviewed by another research group member.

### 2.3. Data Extraction and Synthesis

Data from studies selected for full paper review, including study design, population and sample size, and vaccines and the equity stratifiers studied, were extracted in Excel using a pre-developed data extraction template (see Appendix A). Study authors were approached for further data where necessary.

A meta-analysis was likely to have been unreliable as the studies originated from countries with a wide range of health systems that use different effect measures and equity stratifiers. Consequently, we undertook a narrative review using the Synthesis Without Meta-analysis (SWIM) reporting guidelines [12]. Standard measures of effect (frequencies with 95% confidence interval (CI), unadjusted and adjusted risk and odds ratios (ARR/AOR)) were extracted for each study (where reported). In the presentation and interpretation of the results of the narrative synthesis, the risk of bias assessment and the social, economic and cultural differences between studied populations were considered.

We sought data on the following five outcomes of interest related to routine childhood vaccination coverage during the COVID-19 pandemic: Outcome 1—change in vaccine coverage by equity stratifiers during the pandemic; Outcome 2—change in inequity in vaccine coverage during the pandemic compared with pre-pandemic period; Outcome 3—differences in vaccine-specific inequity in coverage; Outcome 4—differences in age-groups’ specific inequity in coverage; and Outcome 5—differences in inequity in coverage between HICs and LMICs. The synthesis did not include data by region for which no comparative socioeconomic information was reported. Data on outcomes were sought in all included papers.

### 2.4. Risk of Bias Assessment

Each full-text article that met the criteria for inclusion was independently assessed for risk of bias by two reviewers using the Newcastle-Ottawa Quality Assessment Tool for Cohort Studies [13]. This tool includes the assessment of sample selection (representativeness; exposed and non-exposed from the same population; method of ascertainment of a social group; changes in coverage during a pandemic), compatibility (adjustment for confounding variables) and outcome (method of ascertainment of coverage; adequate period of follow up; adequacy of study population followed up). Two researchers (NH and NS) recorded their assessment independently in Excel. Stars (*) indicate the quality requirements for individual components of the domains have been met, i.e., a higher number of stars = lower risk of bias. Differences were resolved by consensus.

## 3. Results

After deduplication, 1453 studies were identified, of which 91 were screened for full paper review, and 13 met the inclusion criteria. Details of the identification and selection of studies are shown in a PRISMA flow diagram (Figure 1). The study from Karachi [10] was excluded as the reported data were incorporated in the study covering the whole Sindh province [14], of which Karachi is the capital city.

The authors sought additional information related to equity stratifiers for five studies [15,16,17,18,19]. Additional data were received from the authors of two studies [15,16], both of which were included in the review. Three studies for which no response was received from authors were excluded due to inadequate data on equity stratifiers [17,18] or failure to distinguish coverage by equity stratifiers collected in the pandemic lockdown period from the pre-pandemic period [19].

### 3.1. Characteristics of Included Studies

Table 1 shows the background characteristics of the included studies, and Table 2 presents the impact of equity stratifiers on childhood vaccination coverage. All the included studies reported on the COVID lockdown in their respective countries during the first wave between March and September 2020. Six studies reported data for a pre-lockdown period in addition to a lockdown period [14,20,21,22,23,24].

The included studies used several approaches to study inequities in childhood vaccination inequities. Country-level income categories as defined by the World Bank (https://data.worldbank.org/ accessed on 3 May 2022) to study variations in routine vaccination rates were reported for child populations in two studies for low-income countries (studies from Ethiopia [25,26]), three for lower-middle-income countries (studies from India [20] and Pakistan [14,16]), three for upper-middle-income countries (studies from Brazil [27], China [28] and Colombia [21]), four for high-income countries (three US studies [22,23,24] and UK [15]) and in a study comparing nine middle-income countries with 16 high-income countries [29].

Approaches for data collection varied across the studies. In four of them, the data collection was performed using a structured interview plus vaccination card records [16,20,25,27], four studies used a structured interview/survey only [15,26,28,29] and five studies used administrative records [14,21,22,23,24].

Reported age groups differed in the selected studies. Children under two years of age were studied in eight studies [14,15,16,20,24,25,26,27], children and adolescents aged 0–18 years were reviewed in two studies [22,29] and children aged 3–17 years were included in the remaining study [28].

The included studies used seven categories of equity stratifiers, which are summarised in Figure 2. Coverage change during the pandemic was reported for the education level of the responding parent in six studies [14,16,20,25,26,28]. Race/ethnicity was also reported [16,22,23], and employment [29] and wealth were measured at the country level based on World Bank criteria [29]. Further, the household level assessed by assets and building characteristics of the home [27] was used as an equity stratifier; of particular interest was the use of the Hindu system of low assets or low caste with its rigid hierarchical groups based on work and religion [20]. Rural versus urban dwelling was also used in two studies [14,21] and area-level measure of slum dwellings in one [14]. For example, in Pakistan and Colombia, rural areas are poorer than urban areas. In a study from the US, registration with Medicaid was used as an equity stratifier available only to low-income families [24].

All included studies reported data on coverage by one or more equity stratifiers; however, the populations studied, data collection methods employed, and equity stratifiers varied widely, making a meta-analysis unreliable (see Table 1 and Table 2).

**Table 1 vaccines-10-01013-t001:** Background characteristics of the included studies.

Study	Year	Country	Study Design	Data Collection Months	Total Number of Children Studied	Age of Children	Sex: Males No. (%)
Ackerson et al. [23]	2021	Southern California, USA	Retrospective cohort study using electronic health records (EHR) of the Kaiser Permanente Southern California integrated health care system	January to August 2020 compared with January to August 2019 2020 data sub-divided into: Pre-pandemic: 1 January–12 March Stay-at-home: 413 March–May 6 Reopening: 7 May to 31 August	987,544 eligible for vaccination on 1 January 2019 992,971 eligible for vaccination on 1 January 2020	0–18 years	504,456 (51.1%) 2019 507,361 (51.1%) 2020
Bell et al. [15]	2021	UK	Mixed methods study with an online cross-sectional survey and semi-structured telephone interviews	19 April and 11 May 2020	1252 parents or guardians of eligible children	18 months or less	Not stated
Bramer et al. [24]	2020	Michigan State, USA	Secondary analysis of routinely collected Michigan State-level data on children eligible for vaccination	May 2020 (pandemic months) compared to May 2019 (pre-pandemic month)	9539 children from the pandemic month and 9269 children from the pre-pandemic month)	1–24 months	Not stated
Chandir et al. [14]	2020	Sindh Province, Pakistan	Secondary analysis of regional electronic data from the Government of Sindh’s Zindagi Mehfooz (Safe Life) Electronic Immunization Registry (ZM EIR)	23 September 2019–22 March 2020 (pre-lockdown period) and 23 March–9 May 2020 (COVID-19 lockdown period)	786,325 children enrolled in pre-lockdown period and 83,360 during lockdown period	0–24 months	407,410 (51.8%) pre-pandemic; 43,728 (52.5%) in lockdown
DeSilva et al. [22]	2021	California, Oregon, Washington, Colorado, Minnesota, and Wisconsin, USA	Surveillance study using a pre-pandemic, post-pandemic control design	February 2020 (Pre-Pandemic); May 2020 (Pandemic restrictions); September 2020 (Post-Pandemic) Same months in 2019 used as controls	39,113 children in 2019 and 40,373 in 2020	Children reaching specified ages (7 months, 18 months, 6 years, 13 years, and 18 years) in February, May, or September 2019 and 2020	1429,979 (51.0%)
Hou Z et al. [28]	2020	Wuhan and Shanghai, China	Cross-sectional online survey	12–17 March 2020	1655 children and young people enrolled in the survey–626 had scheduled vaccinations during the study period	3 to 17 years	830 (50.2%)
Jain et al. [20]	2021	Rajasthan, India	Retrospective observational study based on phone survey–	March 2020 (pre-lockdown);March-May2020 (Lockdown period) and June-July 2020 (post-lockdown period)	2114 children: 443 > 12 months of age before March 2020 (unexposed to lockdown); 722 turned 12 months between March and May 2020 (partially exposed); 796 aged 9 months March to May (heavily exposed); 183 aged 9 months in June-July (post-exposed))	Children born in or after January 2019 and at least 12 months of age at the time of the survey	1122 (52%)
Miretu et al. [25]	2021	Dessie Town, Northeast Ethiopia	A community-based cross-sectional survey using multistage cluster sampling	22 July to 7 August 2020	610 mothers with children aged 15–23 months enrolled.	15–23 months	300 (49.2%)
Moreno-Montoya et al. [21]	2021	Colombia	An ecological study of monthly vaccination data from the Expanded Program of Immunization (EPI)	March to October 2020 (lockdown period March to 1 September) compared with March to October 2019	2,128,642 children in 2019 2,110,767 children in 2020	Three age cohorts: <12 months 12–24 months 5 years	Not stated
Rizwan et al. [16]	2021	Pakistan	Cross-sectional survey	25 July to 7 August 2020	345 children whose parents completed the questionnaire and had up-to-date vaccination cards	<2 years	181 (52.4%)
Shapiro et al. [29]	2021	9 middle- and 16 high-income countries	National panel survey data	14 May to 9 June 2020	9359 children in 9 middle-income countries and 14,886 in 16 high-income countries/	0–17 years	Not stated
Silveira MF et al. [27]	2021	Brazil	Cross-sectional survey	24–27 August 2020	2530 children (vaccination data collected by questionnaire for 2439 children and from vaccination cards for 1547 children)	<2 years	1305 (51.6%)
Tegegne W et al. [26]	2020	Southwest Ethiopia	Cross-sectional mixed-methods survey	2 September to 21 October 2020	1300 children	10–23 months	Not stated

**Table 2 vaccines-10-01013-t002:** Impact of equity stratifiers on childhood vaccine coverage.

Study	Routine Childhood Vaccine Studied	Equity Stratifiers Measured	Analysis	Main Results
Ackerson et al. [23]	HepB, ROTA, DTaP, Hib, PCV13, IPV, MMR, VAR, HepA, Tdap, MenACWY, 9vHPV	Race or ethnicity: Hispanic; Non-Hispanic white; Non-Hispanic Black; Non-Hispanic Asian American	Outcome: Total of all routine vaccine doses administered for ages 0–18 years during pre-pandemic (1 January to 12 March), stay-at-home (13 March to 6 May) periods. Analysis by race or ethnicity: Difference-in-Difference (DID) analyses and estimated adjusted percentage differences and 95% CIs using Poisson regression models, adjusting for the percentage difference in vaccine doses administered during the pre-pandemic period	Adjusted % Difference (95% CI) (Stay-at-Home period v. Pre-pandemic): Hispanic: −46.4 (−47.2,−45.7) Non-Hispanic White: −49.1 (−50.1,−48.2) Non-Hispanic Black: −53.4 (−55.1,−51.7) Non-Hispanic Asian American: −41.7 (−43.3,−40.0)
Bell et al. [15]	Recommended vaccines according to the UK schedule [details of schedule not given in paper]	Race or ethnicity: White; Black and Minority Ethnicity (BAME) Annual Household Income: Low (<£35,000); Medium (£35,000–84,999): High (>£85,000) Employment: Working full-time; Working part-time; Homemaker; Student; Unemployed	AOR with 95% CI for overdue vaccinations by race/ethnicity, household income and employment adjusted for each other and the number of children	Overdue vaccinations by ^4^: Race/ethnicity (White = reference): BAME: AOR 2.15 (95% CI 0.72,6.40) Income (Medium = reference): Low: AOR 1.24 (0.63,2.46) High: AOR 0.93 (0.43,1.97) Employment (Full-time = reference): Part-time: AOR 0.93 (0.47,1.85) Homemaker: AOR 2.23(0.97,5.22) [Student and Unemployed–insufficient numbers for analysis]
Bramer et al. [24]	Pentavalent (HepB+Hib+DTaP), Hexavalent (HepB+Hib+DTaP+IPV), MMR/MR/measles, HepB (separate shot), ROTA	Medicaid enrolment v. non-Medicaid enrolment	Frequencies: up-to-date with vaccinations in May 2020 cf. 2016–19 No probability or analysis of statistical significance for equity stratifiers	Reductions in all age cohorts apart from those aged under one month. Up-to-date series coverage for each age cohort assessed in May 2020 is lower for Medicaid enrolled children than non-Medicaid enrolled children. The largest difference in the 7-month cohort–34.6% Medicaid v. 55.0% non-Medicaid up to date
Chandir et al. [14]	Pentavalent (HepB+Hib+DTaP), Measles, ROTA, PCV, BCG, IPV/OPV	Individual level: Rural/Urban dweller; Maternal Education in years–0,1–8,9–10,11–12,13+. Area level (Union Councils–smallest admin areas): Rural v. Urban; Slum v. Non-slum; Super High-Risk Union Councils (SHRUC)	Analysis 1: age-appropriate Penta 3 vaccination completion during lockdown: ARRs calculated by rural v. urban dweller and maternal education level adjusted for child’s sex, birth in a hospital, Penta 2 vaccination by outreach, outreach vaccination history, age at Penta 2 and BCG. Both rural/urban dweller and maternal education were included in the regression model Analysis 2: area-based decline in average daily vaccine visits lockdown v. baseline Analysis 3: Percentage difference in enrolment COVID-19 lockdown v. baseline by maternal education (95% CI)	Analysis 1: by years of maternal education (no years as ref*): 1–5 years ARR 1.19 (95% CI 1.13,1.25); 6–10 years ARR 1.38 (1.27,1.50); 11–12 years ARR 1.50 (1.35,1.65); 12+ years ARR 1.38 (1.23,1.55). By rural v. urban (rural as ref): ARR 1.03 (0.98,1.08) Analysis 2: [no measure of probability–only frequencies]: % reduction in vaccination visits–Rural–54.9% v. Urban −53.8%; Slum–53.8% v. Non-Slum–51.3%; SHRUC −68.1% v. Non-SHRUC −50.6% Analysis 3: % difference in enrolment by maternal education: none–3.1% (−3.45,−2.74); 1–8 years–0.5% (−0.86,−0.14); 9–10 years–1.3% (1.14,1.45); 11–12 years −1.12% (1.08,1.32); >12 yrs–1.2% (1.01,1.32).
DeSilva et al. [22]	HepB, ROTA, DTaP, Hib, PCV13, IPV, MMR, VAR, HPV, MCV4	Race or ethnicity: Asian; Black; Hispanic; White; Other	Outcome: Up-to-date (UTD) with scheduled vaccines Unadjusted proportions (95% CIs) UTD in May 2020 compared with February 2020 by race/ethnicity stratified by age group	February/May 7 months Asian 0.88 (0.87,0.90)/0.81(0.79,0.83) Black 0.68 (0.63,0.73)/0.54(0.49,0.60) H’panic0.82 (0.80,0.84)/0.71(0.69,0.74) Other 0.80 (0.77,0.82)/0.70(0.67,0.72) White 0.81 (0.79,0.82)/0.72(0.70,0.74) 18 months: Asian 0.78 (0.76,0.80)/0.76 (0.74,0.79) Black 0.49 (0.45,0.53)/0.41 (0.36,0.45) H’panic0.59 (0.58,0.61)/0.56 (0.54,0.58) Other 0.59 (0.57,0.62)/0.54 (0.51,0.57) White 0.59 (0.57,0.61)/0.52 (0.50,0.54) 6 years: Asian 0.81 (0.79,0.83)/0.80 (0.78,0.82) Black 0.74 (0.70,0.77)/0.76 (0.73,0.80) H’panic0.81 (0.79,0.82)/0.80 (0.79,0.82) Other 0.74 (0.72,0.77)/0.72 (0.69,0.74) White 0.79 (0.78,0.81)/0.80 (0.79,0.82) 13 years: Asian 0.71 (0.68,0.74)/0.72 (0.69,0.74) Black 0.54 (0.50,0.58)/0.52 (0.49,0.56) H’panic0.64 (0.63,0.66)/0.64 (0.62,0.66) Other 0.55 (0.52,0.58)/0.50 (0.47,0.53) White 0.53 (0.51,0.55)/0.55 (0.53,0.56) 18 years: Asian 0.63 (0.60,0.67)/0.65 (0.62,0.68) Black 0.60 (0.56,0.64)/0.62 (0.58,0.65) H’panic0.68 (0.66,0.70)/0.67 (0.65,0.69) Other 0.51 (0.48,0.54)/0.49 (0.47,0.52) White 0.56 (0.54,0.58)/0.54 (0.52,0.56)
Hou Z et al. [28]	Scheduled childhood vaccination (excluding COVID vaccine). in children aged 3 to 17 years [details of schedule not given in paper]	Education level of responding parent: High school or below; Some college; Bachelor’s degree or above	OR with 95% CI of delay in vaccination schedule by parent educational level adjusted for city, child’s age and gender, household size, father respondent, COVID−19 cases in the neighbourhood	Delayed scheduled vaccination by educational status (Bachelor’s degree or above as a reference): Some college: AOR 0.75 (95% CI 0.46,1.21); High school or below: AOR 0.30 (0.15,0.59)
Jain et al. [20]	Pentavalent (HepB+Hib+DTaP), Measles, BCG	Low assets, low caste, and low parent education	Analyses of interest: To study changes in the percentage of children fully immunised during lockdown compared with prior to lockdown (data only for the heavily exposed group (796) and unexposed group (443)): Analysis 1: Percentage point (PP) differences in fully immunised rates between exposed groups by equity stratifiers (95% CIs) Analysis 2: PP differences in equity stratifier difference (95% CI)	Assets: Analysis 1: High (77.1–70.4) PP = −6.5 (95% CI −13.03,−0.30); Low (71.5–59.3) PP = −12.4 (−19.9,−4·8) Analysis 2: PP difference between exposed groups by asset level difference = −5.6 (−15.8,4.6) Caste: Analysis 1: High (74.8–66.8) PP = −8.4(−13.9,−2.2); Low (72.6–52.9) PP = −13.5 (−23.5,−3.0) Analysis 2: PP difference between exposed groups by caste difference = −5.6 (−17.7,6.4). Parent education: Analysis 1: High (79.0–73.2) PP = −5.8(−12.6,1.1); Low (70.1–57.1) PP = −13.0 (−20.3, −5.7) Analysis 2: PP difference between exposed groups by education difference = −7.2 (−17.3,2.8)
Miretu et al. [25]	Pentavalent (HepB+Hib+DTaP), PCV, Measles, ROTA, BCG, OPV	Education status of mother/caregiver: Cannot read or write No schooling but can read or write School grades 1–8 School grade 9–12 College/University or above	Outcome: Fully vaccinated Logistic regression: Odds ratio (OR) of being fully vaccinated in different educational statuses compared to mothers/caregivers who cannot read or write (reference group) by education status adjusted for marital status, father as the main caregiver, and distance to a health facility	Can read or write: AOR 7.82 (95% CI 1.24,49.2); Grade 1–8: AOR 5.23 (1.23,20.22); Grade 9–12: AOR 2.71 (0.65,11.25); College/University: AOR 3.91(0.92,16.60)
Moreno-Montoya et al.[21]	BCG, HepB, IPV, OPV, ROTA, PCV, Pentavalent (HepB+Hib+DTaP), Hib, MMR, VAR, HepA, YF.	Rural v. urban areas	Two-level multilevel linear regression model to assess the effect of rural residence on absolute differences in individual vaccine coverage at different ages between 2019 and 2020; the geographical area was considered a level 2 variable, and rural residence a level 1 variable. The effect size of rural residence expressed as Beta coefficients (95% CI)	Beta coefficients by rural residence: <12 months: BCG 0.45 (0.22,0.69); Hep B 0.49 (0.25,0.73); IPV 0.36 (0.16,0.57); OPV+IPV 0.26 (0.06,0.48); Penta ^1^ 0.39 (0.19,0.58); Rota ^2^ 0.37 (0.18,0.56); Pneumococcal ^3^ 0.39 (0.19,0.59) 12–23 months: MMR 0.14 (−0.05,0.33); Varicella 0.21 (−0.01,0.41); Pneumococcal 0.17 (−0.03,0.37); Hep A 0.16 (−0.04,0.37); Yellow Fever 0.12 (−0.08,0.33); OPV 0.10 (−0.08,0.29); DPT 0.13 (−0.06,0.31) 5 years: MMR 0.25 (0.06,0.45); OPV 0.22 (0.02,0.43); DPT 0.28 (0.08,0.49)
Rizwan et al. [16]	Pentavalent (HepB+Hib+DTaP), MMR, BCG, OPV	The educational level of mother and father Illiterate; Primary; Secondary; Graduate; Masters; Professional degree Monthly income in rupees (<20,000; 20,000–50,000; 50,000–100,000; >100,000)	Outcome: any missed vaccination during the pandemic Univariate analysis with chi-square and *p* value	Frequencies ^4^ with *p* value Father’s education: Illiterate 40.7%; Primary 51.2%; Secondary 32.7%; Graduate 40.3%; Masters 0%; Professional degrees 38.5% *p* = 0.006 Mother’s education: Illiterate 42.7%; Primary 76.3%; Secondary 31.2%; Graduate 25.5%; Masters 0%; Professional degrees 71.4% *p* = 0.082 Monthly income ^5^: low (not specified) *p* = 0.023
Shapiro et al. [29]	Routine childhood vaccinations in country vaccination schedules	Individual respondent (not stated if head of household): Employment status (unemployed, retired, student v. employed full or part-time) Transnational level: Income level of a country defined by World Bank: Middle- v. High-income	Outcome: Any missed or delayed vaccinations during the pandemic. Individual level: OR by employment status adjusted for COVID-19 risk factors, respondent’s sex, age, psychological distress, household size, and the number of children. Transnational level: Frequencies by middle-and high-income countries compared by *t*-tests (*p* values)	Missed childhood vaccinations: By employment status (not working reference) stratified by country income group: Working (Middle-income countries) AOR 1.38 (95% CI 1.14,1.67). Working (High-income countries) AOR 1.75 (1.36,2.25) Transnational level: Middle income countries 7.6% v. High-income countries 3.0% (*p* value < 0.05)
Silveira MF et al. [27]	Pentavalent (HepB+Hib+DTaP), MMR, HepB, BCG, OPV	Household wealth quintiles (based on household assets and characteristics of the building)	The proportion (95% CI) of children with any missed dose of scheduled vaccines under 3 years of age (schedule of vaccinations in 2nd year not specified) was analysed from questionnaire responses and vaccination card records by wealth quintiles	Missed vaccination–questionnaire responses: Wealth quintile: Q1(poorest) 22.5%(95% CI 19.3,26.2); Q2 21.0%(17.7,24.7); Q3 17.0%(13.9,20.6); Q4 17.5%(14.0,21.6); Q5 (wealthiest) 15.0%(11.6,19.1) *p* for linear trend = 0.03. Vaccination card record: Wealth quintile: Q1 24.5%(20.6,28.8); Q2 24.5%(20.2,29.4); Q3 19.0%(15.0,23.7); Q4 17.8%(13.6,23.0); Q5 15.6%(11.3,21.1) *p* for linear trend = 0.01.
Tegegne W et al. [26]	Pentavalent (HepB+Hib+DTaP), Measles, ROTA, PCV, BCG, IPV/OPV	Maternal Education: Cannot read or write Able to read or write Primary and Secondary SchoolDiploma, Degree and above	OR with 95% CI for incomplete vaccination (defined as a child who missed at least one dose of the included vaccines) by maternal education adjusted for marital status, place of delivery, waiting time at a health facility, means of transportation to a health facility	Incomplete vaccination by maternal education (Diploma, degree or above as a reference): Can not read or write: AOR 5.08 (95% CI 2.31,11.14); Can read or write: AOR 3.46 (1.31,12.86); Primary and Secondary school: AOR 3.54(1.59,7.89)

^1^ Penta–2nd dose; ^2^ Rota–1st dose; ^3^ Pneumococcal–2nd dose; ^4^ obtained on request from authors; ^5^ frequencies by income not provided by authors. OR = odds ratio; AOR = adjusted odds ratio; RR = risk ratio; ARR = adjusted risk ratio; BCG = Bacille Calmette–Guerrin (Tuberculosis) Vaccine, DTaP = Diphtheria and Tetanus toxoids and acellular Pertussis vaccine, paediatric formula, HepA = Hepatitis A Vaccine, HepB = Hepatitis B Vaccine, Hib = Haemophilus Influenzae type b vaccine, 9vHPV = 9-Valent Human Papilloma Virus Vaccine, Men-ACWY = Meningococcal Conjugate Vaccine, quadrivalent, MMR = Measles, Mumps, Rubella Vaccine, PCV/PCV13 = Pneumococcal Conjugate Vaccine (13-valent), IPV = Inactivated Poliovirus Vaccine, OPV = Oral Poliovirus Vaccine, ROTA = Rotavirus Vaccine, Tdap = Tetanus, diphtheria and acellular pertussis vaccines, adult/adolescent formulation, VAR = Varicella Vaccine, YF = Yellow Fever.

### 3.2. Risk of Bias Assessment

Table 3 details the risk of bias assessment of the included studies by the Newcastle-Ottawa scale [13]. The risk of bias was lowest for three of the included studies [14,25,27], with maximum stars for all three scale domains (8 stars). Eight studies [15,20,21,22,24,26,28,29] had a moderate risk of bias (7 or 6 stars) and two studies [16,23] had a high risk of bias (<6 stars).

#### A Narrative Synthesis of Outcomes

In the following, a narrative synthesis of the five different outcomes of this systematic review is provided.

### 3.3. Narrative Synthesis of Outcome 1: Change in Routine Childhood Vaccination during the COVID-19 Pandemic by Equity Stratifiers

Seven studies [15,16,25,26,27,28,29] reported coverage by the equity stratifiers during the pandemic period alone. A single study [14] reported separate analyses for the pandemic period alone and a comparison of pre-pandemic with pandemic coverage (see narrative synthesis of outcome 2).

#### 3.3.1. Parental Education

Chandir et al. [14] reported a social gradient for receiving the age-appropriate third dose of Pentavalent vaccine by maternal education. The authors used the lowest maternal education category, ‘no years of education’, as the reference category. Contrary to the usual convention, an adjusted relative risk (ARR) greater than unity represented an increased likelihood of age-appropriate completion of the third dose of the Pentavalent vaccine. The ARR by the highest maternal education category (12+ years of schooling) was 1.38 (95% CI 1.23–1.55), adjusted for child’s sex, place of birth, Penta 2 vaccination by outreach, outreach vaccination history, age at Penta 2 and age at Bacille Calmette-Guérin (BCG) and rural dwelling.

Completion of the vaccine schedule (see Table 2) by maternal education was reported by Miretu et al. [25]. Although AORs for all maternal education categories compared to the reference category (cannot read or write) were greater than 2, confidence intervals in the two highest categories (Grade 9–12 and College/University) crossed unity.

Tegegne et al. [26] also reported a social gradient in incomplete vaccination by maternal education level. An adjustment for marital status, place of delivery, waiting time at health facility and means of transportation to health facility increased the gradient. Compared with the highest maternal education category (diploma, degree or above), the AORs for complete vaccination were greater than 2 in all other education categories.

Rizwan et al. [16] reported a significant association of frequency of missed vaccination during the pandemic lockdown period in Pakistan with fathers’ but not mothers’ education (*p* values 0.006 and 0.082, respectively). The authors stated that the chi-square test was employed to estimate the *p* values but did not specify whether the estimates were for a linear trend.

The study from Wuhan and Shanghai in China [28], based on a survey of households with children aged 3–17, reported delayed vaccination during lockdown. It found a reverse positive social gradient with less educated parents less likely to have delayed vaccination than more educated parents (AOR 0.30 (0.15,0.59)).

#### 3.3.2. Other Equity Stratifiers

Rural dwellers in Sindh, Pakistan [14] were less likely to have received the age-appropriate Pentavalent 3 vaccine during lockdown than urban dwellers; however, the association was insignificant. (ARR 1.03, 95% CI 0.98,1.08).

Silveira et al. [27] reported a social gradient in the proportion (95% CI) of missed vaccinations based on parental reports and vaccination cards by wealth quintile. According to parental report, the proportions were 22.5% (95% CI 19.3, 26.2) of the children in the poorest quintile compared with 15.0% (11.6, 19.1) of the children from the wealthiest quintile and according to vaccination card record, the percentages were 24.5% (20.6, 28.8) compared with 15.6% for the same respective categories (95% CI 11.3, 21.1).

Low-income households in Rizwan et al.’s study [16] were more likely than higher income households to report missed vaccinations for children <2 years of age (*p* = 0.023). Bell et al. [15] also reported that the likelihood of overdue scheduled childhood vaccinations among low income and black and ethnic minority households was greater (AOR 1.24 and 2.15, respectively), but confidence intervals crossed unity. Shapiro et al. [29] reported on national panel studies that a polling company conducted in nine middle-income countries and sixteen high-income countries. They found that missed routine childhood vaccinations were significantly more likely in both middle-income (AOR 1.38 (1.14, 1.67)) and high-income (AOR 1.78 (1.26, 2.25)) countries in households with respondents in full- or part-time employment compared with unemployed, retired or students.

### 3.4. Narrative Synthesis of Outcome 2: Change in Inequity in Routine Childhood Vaccination Coverage during the COVID-19 Pandemic Compared with Baseline

Six studies [14,20,21,22,23,24] included coverage data during the COVID-19 pandemic lockdown compared with a previous period. Chandir et al. [14] studied the percentage difference in enrolment in Sindh Province, Pakistan, during the COVID-19 lockdown (23 March to 9 May 2020) compared with the pre-lockdown period (23 September 2019, to 22 March 2020) by maternal education. They found a statistically significant difference in the percentage of enrolment between mothers with low education and those with higher education (Table 2). They also reported a percentage reduction in vaccination visits during the lockdown period compared with the baseline by area-based equity stratifiers. Visits reduced in rural areas by 54.9% compared with the 47.5% reduction in urban areas, in slum areas by 53.8% compared with 51.3% in non-slum areas and by 68.1% in Special High-Risk Union Councils (SHRUC), designated as polio-endemic super high-risk sub-districts, compared with 50.6% in non-SHRUCs. No probability measures were reported with these findings.

Jain et al. [20] reported the results of a study conducted in Rajasthan, India. They compared percentage point (PP) differences (95% CI) in fully vaccinated rates in the group reaching nine months of age during the COVID-19 lockdown (March to May 2020) with the group aged >12 months before March 2020. In this comparison, they used three equity stratifiers, i.e., assets, caste, and parental education (Table 2). There were significant reductions in fully vaccinated rates in both high and low caste groups but only in low assets and low parental education groups. When PP differences between the group exposed to the lockdown period and those not exposed were examined by high v. low levels for each equity stratifier, there were no significant differences.

Moreno-Montoya et al. [21], based on Colombian national administrative vaccination data, reported a more significant reduction in the coverage of individual vaccines in rural versus urban areas in the pandemic lockdown period (March to September 2020) compared with the pre-pandemic period (March to October 2019). Reductions in all scheduled vaccines were significantly higher in rural areas than urban ones among children aged <12 months and those aged 5 years. No significant differences were found among children 12–23 months of age (Table 2).

Bramer et al. [24] reported a reduction in Michigan state, USA, in routine vaccination enrolment in all age cohorts (1, 3, 5, 7, 16, 19 and 24 months) in May 2020 compared with May in the preceding three years 2016–2019 (Table 2). Up-to-date coverage of age-appropriate vaccines decreased in all age cohorts, except the cohort of infants aged less than one month. The coverage for each age cohort assessed in May 2020 was lower for Medicaid enrolled children than non-Medicaid enrolled children. The largest difference was found in the 7-month cohort, with 34.6% of Medicaid recipients compared with 55.0% non-Medicaid recipients up-to-date with age-appropriate vaccines (data for other age cohorts by Medicaid receipt was not reported in the paper). However, a statistical analysis of equity stratifiers was not reported.

Ackerson [23] reported on the percentage reduction in total routine vaccine doses administered to Californian children from 0 to 18 years in the ‘stay-at-home’ period of the pandemic (13 March to 6 May 2020) compared with the pre-pandemic period (1 January to 12 March 2020). The results show a significant gradient in the percentage reduction by race/ethnicity. The reduction was most significant among Non-Hispanic Blacks, followed by Non-Hispanic Whites and Hispanics, with the smallest reduction among Non-Hispanic Asian Americans (Table 2).

DeSilva et al. [22] reported the results of a study on eight health systems in six US states. They compared the rates of up-to-date scheduled vaccinations in February 2020 (pre-pandemic) with May 2020 (pandemic period) by race/ethnicity in five age groups (7 months, 18 months, 6 years, 13 years, and 18 years). Up-to-date vaccination rates at 7 months of age reduced significantly in all race/ethnic groups (see Table 2). They reported that at 18 months, non-significant reductions in rates were found in all race/ethnic groups, and, in the older age groups, there were no significant rate reductions. Rates among Black children aged 7 and 18 months were lower than those in other race/ethnic groups during both the pre-pandemic and pandemic periods.

### 3.5. Narrative Synthesis of Outcome 3: Differential Coverage for Specific Vaccines during the COVID-19 Pandemic by Equity Stratifiers

Two studies reported on the coverage of specific vaccines by equity stratifiers [14,21]. Only the study by Moreno-Montoya et al. [21] compared inequity in coverage rates for more than one specific vaccine. Effect sizes of absolute differences in pandemic and pre-pandemic coverage of individual vaccines were expressed as Beta coefficients (95% CI). They found that the difference between rural and urban areas varied in children <12 months of age, with the highest difference for BCG (OR 0.45 (0.22, 0.69)) and the lowest for Polio Vaccine (OPV or IPV) (OR 0.26 (0.06, 0.48)). However, absolute differences in coverage did not reach statistical significance at the 95% level for any of the individual vaccines in the three age groups studied.

### 3.6. Narrative Synthesis of Outcome 4: Differential Coverage in Age Groups during the COVID-19 Pandemic by Equity Stratifiers

DeSilva et al. [22], as reported above, showed inequity by race/ethnicity on up-to-date vaccinations during the pandemic among children at 7 and 18 months of age but not among older age groups (6, 13 and 18 years). Effect sizes of absolute differences during the pandemic compared with pre-pandemic coverage rates for scheduled vaccines by rural versus urban dwellings in Colombia [21] were higher among children <12 months of age than those aged 12–24 months and those aged 5 years.

Bramer et al. [24] reported higher rates of coverage reduction in May 2020 compared with May 2016–19 among Medicaid enrolled compared with non-Medicaid enrolled children in all age cohorts, except for those one month of age, and the difference in the rate reduction was highest in the 7-month cohort. The remaining studies do not report comparative vaccination coverage data by age group. The studies reporting on children 0–2 years of age all report social gradients in vaccination coverage in the same direction.

### 3.7. Narrative Synthesis of Outcome 5: Inequity in Coverage between HICs and LMICs during the Pandemic

Households in MICs polled in Shapiro et al.’s study [29] were more likely than those in HICs to report missed routine childhood vaccinations during the pandemic (7.6% v. 3.0%; *p* < 0.05). Their study is the only one to directly compare vaccination coverage between HICs and LMICs during the pandemic. A direct comparison of the magnitude of inequity between HICs and LMICs in the remaining included studies cannot be made due to the wide variety of equity stratifiers used. However, inequity in routine vaccination coverage was reported in all four country income groups as defined by the World Bank (low-, lower-middle-, upper-middle- and high-income countries).

### 3.8. Outcome 1: Change in Routine Childhood Vaccination Coverage during the COVID-19 Pandemic by Equity Stratifiers

Of the studies reporting on routine childhood vaccination coverage rates during the pandemic period alone or separately from the pre-pandemic period, six [14,15,16,25,26,27] reported lower rates among children in disadvantaged population groups. Based on the risk of bias assessments (see above and Table 3), the strongest evidence of inequity among the included studies comes from Chandir et al.’s low-risk-of-bias study [14]. Although the remaining five studies [15,16,25,26,27] reported inequity to be consistent with Chandir et al.’s [14] findings, the strength of the evidence is limited by the higher risk of bias, from moderate [15,25,26] to high [16,23].

Contrary to the findings from the above studies, the studies of Hou et al. [24] and Shapiro [29], both with a moderate risk of bias, reported higher rates of missed or delayed vaccination schedules during the pandemic among children from more advantaged households measured by education [24] and employment status [29].

Studies comparing coverage in pre-pandemic and pandemic periods are considered in the next section; however, all six studies, with low [14,21,23] and moderate [20,22,24] risk of bias, show evidence of inequity in coverage reduction during the pandemic.

In summary, the strength of the evidence for inequity in vaccination coverage during the pandemic lockdown is moderate.

### 3.9. Outcome 2: Change in Inequity in Routine Childhood Vaccination Coverage during the COVID-19 Pandemic Compared with Baseline

Three low-risk-of-bias studies reported evidence of increased inequity during the pandemic lockdown months compared with pre-pandemic months [14,21,23]. Robust evidence of increased inequity between rural compared with urban dwellers in individual vaccine coverage during the pandemic period was reported by Moreno-Montoya et al. [21]. Ackerson et al. [23] also reported robust evidence of increased inequity by race/ethnicity during the pandemic compared with the pre-pandemic period. Chandir et al. [14] showed differences by education and by area-level equity stratifiers in a reduction in vaccine enrolment during the pandemic compared with baseline; however, the evidence was less robust as the analysis was unadjusted for confounding effects.

Two moderate risk-of-bias studies [22,24] provided weak evidence of an increase in inequity in the pandemic compared with the pre-pandemic period and the same month in the three preceding years. An increase in inequity in up-to-date vaccinations in May 2020 compared with the same month in the three preceding years was reported by Bramer et al. [24]; however, percentage reductions were reported for a single age cohort but without probability measures or adjustment for confounding. DeSilva et al. [22] compared up-to-date vaccination rates in May 2020 with those in February 2020 by race/ethnicity. Rates were reduced in all ethnic groups among children who were 7-months old. However, in the absence of a comparative analysis of differences in reduction by ethnic group, there was no clear evidence of an increase in inequity.

Jain et al.’s higher-risk-of-bias study [20] reported statistically significant percentage point differences in full immunisation rates during the lockdown months compared with pre-pandemic months by several equity stratifiers. These included separate high and low equity asset, caste, and education categories. However, percentage point differences between high and low equity stratifier categories during the pandemic lockdown months compared to those prior were not statistically significant. The results suggest that the reductions occurred equally across equity stratifiers, and inequity did not increase.

In summary, although two moderate risk-of-bias studies [20,22] failed to show evidence of increasing inequity during the pandemic, three low-risk-of-bias studies [14,21,23] report moderately strong evidence of increasing inequity, with additional, but weak, supporting evidence from Bramer et al. [24].

### 3.10. Outcome 3: Differential Coverage for Individual Vaccines during the COVID-19 Pandemic by Equity Stratifiers

We identified one low-risk-of-bias study [21], which reported coverage of individual vaccines by equity stratifiers during the pandemic.

In summary, evidence for differential inequity is not supported as differences for individual vaccines by rural versus urban dwellings failed to reach statistical significance.

### 3.11. Outcome 4: Differential Coverage in Age Groups during the COVID-19 Pandemic by Equity Stratifiers

Coverage rates during the pandemic in different age groups by equity stratifiers are reported by one low [21] and two moderate [22,24] risk-of-bias studies. The evidence suggests that inequity during the pandemic was more likely to affect routine vaccination in children of less than 12 months of age.

In summary, the small number of studies reporting inequity by age group and differences in the age groups studied make this evidence inconclusive.

### 3.12. Outcome 5: Inequity in Coverage between HICs and LMICs during the Pandemic

We identified a single moderate risk of bias study [29] reporting empirical evidence on inequity in routine childhood vaccination coverage during the pandemic among families in middle- compared with high-income countries. The study shows higher rates of missed childhood vaccinations in middle-income countries.

In summary, data from a single study represent weak evidence for this outcome.

## 4. Discussion

Our systematic review sought evidence of the impact of measures to control the spread of the SARS-CoV-2, such as lockdowns, on routine childhood vaccination coverage inequity. Our literature search from 1 January 2020 to 18 January 2022 identified 13 studies with data on coverage by equity stratifiers. By means of a narrative synthesis, we found moderate evidence for inequity in reducing the vaccination coverage of children and adolescents during lockdowns between March and September 2020 and moderately strong evidence for an increase in inequity compared with pre-pandemic months. While the review found evidence of greater inequity in coverage among children less than 18 months old compared with other age groups and inequity between MICs and HICs, there were too few studies to judge the strength of the evidence, which remains inconclusive.

Inequity is a crucial driver of sub-optimal levels of routine vaccination coverage [4,30]. The small number of studies from nine individual countries and a group of 25 countries published in the first 21 months of the pandemic suggests limited research on inequity’s contribution to the reduction in routine vaccination coverage internationally [1,30,31]. WHO [4] estimated the potential for improvement in vaccination coverage by eliminating economic-related inequity in 23 priority study countries (see Figure 3.15, *p*. 35 in the report). It was concluded that all 23 countries would achieve higher national rates by eliminating economic-related inequity. The most significant potential for improvement was reported in Nigeria, where the national rate of DTP3 vaccination coverage would improve by 40 percentage points by eliminating economic-related inequity. Similarly, Ethiopia and Pakistan, two countries represented in our review, would improve national rates by over 20% by eliminating inequity. In addition, target 3.B.1 of the SDGs, access to affordable essential medicines and vaccines for all by 2030, measured by the indicator “proportion of the target population covered by all vaccines included in their national program” [31], will only be achieved by eliminating inequity, including that resulting from the pandemic.

The reduction in coverage among the most vulnerable children in the pandemic, as reported in the two studies from Pakistan included in the review [14,16], is likely to impede its progress in achieving the targets set by the Global Vaccine Action Plan (GAVP) and the Immunization Agenda 2030 [32]. Pakistan was identified in the Global Burden of Disease study [32] as one of the countries with a high level of ‘zero-dose’ children, defined as those not receiving any doses of the diphtheria-tetanus-pertussis vaccine. The pandemic is likely to have added to this number among children of mothers with low levels of education and in low-income households. Similar reductions in Ethiopia, reported in the two studies [25,26] included in the review, are likely to affect countrywide vaccination coverage and GAVP targets. Nonetheless, it is difficult to gauge the probable extent of the reduction as the numbers in the lowest education group (cannot read or write) in both studies are very low.

A Brazilian study [27] reported that over 20% of children in the two poorest quintiles missed vaccinations during the pandemic compared with 17%, 17.5% and 15% in higher income quintiles. If this level of inequity occurred throughout the country, it would likely have made the GAVP targets more difficult to achieve. Access to routine vaccination for children in rural areas in LMICs is known to be more difficult than for those in urban areas [4]. The study from Colombia [21] indicates that the pandemic is likely to have increased accessibility problems for rural families, especially among young children.

Pre-pandemic vaccination coverage rates among Black children and those on Medicaid in the USA [33] were low compared with other ethnic groups and non-Medicaid enrolled children. The studies by Ackerson et al. [23] and Bramer et al. [24] suggest that the vaccination coverage in these groups is likely to have further reduced due to the pandemic increasing existing inequities.

The review has the following strengths. The review protocol was registered with PROSPERO, and the search strategy was designed and conducted by a health sciences librarian. We searched nine electronic databases comprehensively, including MedRxiv, the pre-print database, and the WHO Global Research on Corona Virus (COVID19) database. We included all studies without any language restrictions. In addition, the search was up to date till January 2022. PRISMA 2020 guidelines [34] were adhered to when screening abstracts, selecting studies for full paper review and identifying studies for inclusion (see Appendix A).

Our review has some limitations. We did not undertake a grey literature search and might have missed some data. However, we aimed to include only published data. A meta-analysis was not feasible due to widely differing health systems, outcome and effect size measures and equity stratifiers; however, we followed the standard SWiM guidelines for the narrative synthesis and provided a comprehensive review of the available data [12].

Measures to control the pandemic have disproportionately affected children in poor households and marginalised and isolated communities in LMICs [5] and HICs [6]. They have also disrupted their access to healthcare [35,36]. A UN Development Programme study [37] on the impact of COVID-19 on Sustainable Development Goals found that 44 million people, many of whom will be children, are expected to be pushed into extreme poverty by 2030 due to COVID-19. Applying an equity lens to the study of the impact of the pandemic on routine vaccination coverage is essential to identify children who have missed vaccinations and those with no previous access to vaccination programmes [35]. Our review found moderate evidence of equity-stratifier-specific inequity in routine childhood vaccination coverage during the pandemic and more robust evidence of an increase in equity-stratifier-specific inequity in the pandemic compared with the pre-pandemic period. Evidence of differential inequity by specific vaccines and child age groups was inconclusive. However, the evidence was confined to only thirteen eligible studies from a small number of countries. Before the pandemic, the WHO recommended expanded health-inequality monitoring, especially in low-income countries, to reduce inequity in childhood immunisation and inform equity-oriented programmes aimed at reaching the most disadvantaged population sub-groups [4]. The findings of this review indicate that inequity is likely to have increased due to the pandemic. However, more extensive and robust evidence is needed to accurately identify the extent of the pandemic’s short, medium and long-term effects on childhood vaccination inequity. Methodologically robust research covering a wide range of LMICs and HICs using routine vaccination coverage data is available for many countries [1,32]. These include household surveys, such as Demographic and Health Surveys (DHS) [38], Multiple Indicator Cluster Surveys (MICS) [39], and administrative data collection systems. There is an urgent need to synthesise the current evidence and inform interventions to reverse the pandemic’s effects on childhood vaccination inequity.

## Figures and Tables

**Figure 1 vaccines-10-01013-f001:**
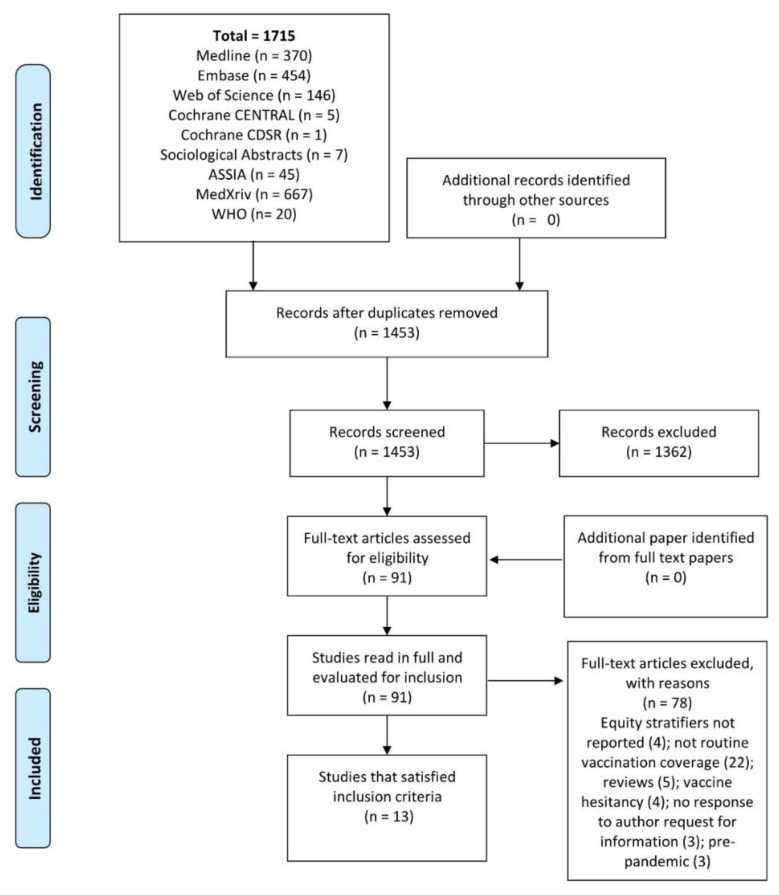
PRISMA flow diagram of article identification retrieval and inclusion.

**Figure 2 vaccines-10-01013-f002:**
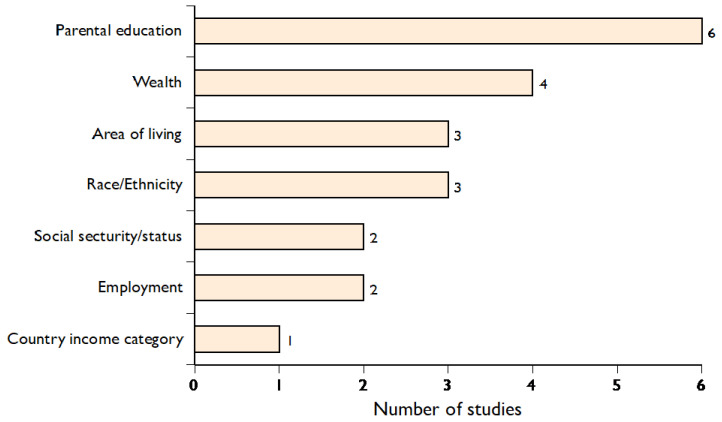
Ranking of equity stratifiers from included studies.

**Table 3 vaccines-10-01013-t003:** Quality assessment (Risk of Bias).

	Selection				Comparability	Outcome			
Author	Representative of Exposed Children (Low Social Group) in the Population: Yes */Partly */No/Not Stated	Non-Exposed Children (Higher Social Group) from the Same Population: Yes */No/Not Stated	Ascertainment of Exposure (Social Group): Administrative Records */Structured Interview */Self-Report/Not Stated	Change in Coverage Data Collected during the Period of the Pandemic Not Before: Yes */No	Study Controls for Potential Confounding Variables: Yes */No	Ascertainment of Vaccine Coverage: Data Linkage */Self-Report/Not Stated	Was the Period of Ascertainment Adequate to Identify a Difference in Coverage: Yes */No/Not Stated	Adequacy of Follow-Up: All Children Accounted For */>70% Accounted For */<70% Acoounted For/Not Stated	No. of Stars
Ackerson et al. [23]	Partly *	Yes *	Administrative records *	Yes *	Yes *	Data Linkage *	Yes *	All *	8
Bell et al. [15]	No	Yes *	Structured interview *	Yes *	Yes *	Self-Report	Yes *	>70% *	6
Bramer et al. [24]	Not Stated	Yes *	Administrative records *	Yes *	No	Data Linkage *	Yes *	All *	6
Chandir et al. [14]	Yes *	Yes *	Administrative records *	Yes *	Yes *	Data Linkage *	Yes *	All *	8
DeSilva et al. [22]	Yes *	Yes *	Administrative records *	Yes *	No	Data Linkage *	Yes *	All *	7
Hou et al. [28]	Partly *	Yes *	Structured interview *	Yes *	Yes *	Self-Report	Yes *	>70% *	7
Jain et al. [20]	Partly *	Yes *	Structured interview *	Yes *	Yes *	Self-Report	Yes *	>70% *	7
Miretu et al. [25]	Partly *	Yes *	Structured interview *	Yes *	Yes *	Self-Report & Vacc Cards	Yes *	All *	7
Moreno-Montoya et al. [21]	Yes *	Yes *	Administrative records *	Yes *	Yes *	Data Linkage *	Yes *	All *	8
Rizwan et al. [16]	No	Yes *	Structured interview *	Yes *	No	Self-Report & Vacc Cards	Yes *	Not Stated	4
Shapiro et al. [29]	Partly *	Yes *	Structured interview *	Yes *	Yes *	Self-Report	Yes *	Not Stated	6
Silveira et al. [27]	Partly *	Yes *	Structured interview*	Yes *	No	Self-Report and Vacc Card	Yes *	<70%	5
Tegegne et al. [26]	Partly *	Yes *	Structured interview *	Yes *	Yes *	Self-Report	Yes *	Not Stated	6

Stars (*) indicate quality requirements for individual components of the domains have been met, i.e., higher number of stars = lower risk of bias.

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
