# Peer review of "The Impact of COVID-19 Pandemic on Inequity in Routine Childhood Vaccination Coverage: A Systematic Review"

_vaccines, 2022, doi:10.3390/vaccines10071013_

Round 1

Reviewer 1 Report

Thanks for the invitation. This is an interesting piece of work where the authors have compiled the data of 13 studies in order to systematically synthesize the evidence to address the existing gap in the body of evidence and assess the impact of the COVID 19 pandemic on inequities in routine childhood vaccination coverage.

Methods: Please make it clear that all study screen, search, selection, inclusion, and exclusion was performed in duplicates. Please describe the outcome variables included in this study. I noticed that the authors used equity stratifiers in the results but there is no information on this classification in the method section. All study outcome variables must be defined and described in the methodology section of this review. I suggest removing the panel of outcomes from the methods and discussing these outcomes in the methodology section of the paper. 

Results: Table 1: please present the table in Landscape format. 

I have the suggestion to make a ranking diagram or Bar chart showing the ranks of equity stratifiers so readers could see which factor was more common in vaccine equity.

The limitations section of the manuscript is not adequately addressed. There is a need to consider the quality of the studies in the limitation section and most of the studies are of low quality. The authors need to make it clear that the current evidence that originated from this review is of medium quality or of low quality. 

The conclusion section is quite general and vague, authors are suggested to please revise this section and provide the major results according to the review question. Also, provide future direction for research and suggestions for policymakers to improve vaccine equity.

Reviewer 2 Report

Due to the inhomogeneity of study design and the wide range of healthcare delivery systems in the countries studied, meta-analysis was not possible. Therefore, the authors undertook a narrative synthesis (?qualitative assessment) using the ‘synthesis without meta-analysis’ (SWIM guidelines). A reference was not given to this SWIM methodology. Tables 1 and 2, provide a descriptive summary of each study, and the outcomes synthesised using this approach were: 

 Outcome 1: Change in routine childhood vaccination coverage during the COVID-19 pandemic by equity stratifies  

Outcome 2: Change in inequity in routine childhood vaccination coverage during the COVID-19 pandemic compared with the pre-pandemic period  

Outcome 3: Differential coverage for specific vaccines during the COVID-19 pandemic by equity stratifiers  

Outcome 4: Differential coverage in age groups during the COVID 19 pandemic by equity stratifiers Outcome 5: Inequity in coverage between HICs and LMICs during the pandemic 

This approach identified, within the constraints imposed by the methodology, evidence for reduction in childhood vaccine coverage, during the pandemic, with the effect being more pronounced in countries with a lower income . 

There was also moderately strong evidence of decrease in equity of vaccine coverage, and there was some evidence that this parameter was more pronounced in the lower income countries. 

The discussion is balanced and concludes that appropriately standardised and well-designed prospective studies across the range of high-, middle- and low-income countries, would be highly desirable to strengthen the evidence on the effectiveness of vaccine coverage globally, and therefore to inform appropriate interventions. 

With all the constraints encountered, this is an important study that could contribute to reconstitution of better childhood routine vaccine coverage as we come out of the pandemic. 

Specific comments to authors: 

I would congratulate the authors for addressing this incredibly important topic that is difficult to evaluate based on the currently available information. 

1. The number of studies included for full evaluation, are referred to in figure 1, the methods section, Tables 1 and 2, and in the abstract which states that 20 studies were evaluated in detail. I counted 26 studies listed in tables 1 and 2. Revision, needs to identify precisely how many studies were evaluated in detail and there needs to be  consistency between the various sections of the paper on the total number evaluated, in full. 

Please clarify and achieve consistency in the different sections of the paper on the total number of studies finally evaluated. 

2. To improve the accessibility of this important study, to a non-specialist audience, it would be useful to explain the basis of the tools used in this study, for example the Newcastle-ottawa method, and the swim method, and what you exactly mean by narrative? Does this mean you're describing certain aspects of the study that you think are important? 

3. Tables 1 and 2 give a descriptive summary of the studies you have evaluated in detail. It would help the reader if you synthesised information in this long table into a more user-friendly summary table. 

4. If you wish this paper to be widely assimilated by non-specialist readers including those who fund healthcare, including governments, it would be important to reflect if the style of the paper could be improved,  to make it easier to read and assimilate. 

Round 2

Reviewer 1 Report

Thank you for addressing the comments.

Reviewer 2 Report

The authors have satisfactorily addressed all the reviewers comments.